POINT OF VIEW

# Managing a sustainable deep-sea 'blue economy' requires knowledge of what actually lives there

**Abstract** Ensuring that the wealth of resources contained in our oceans are managed and developed in a sustainable manner is a priority for the emerging 'blue economy'. However, modern ecosystem-based management approaches do not translate well to regions where we know almost nothing about the individual species found in the ecosystem. Here, we propose a new taxon-focused approach to deep-sea conservation that includes regulatory oversight to set targets for the delivery of taxonomic data. For example, a five-year plan to deliver taxonomic and genomic knowledge on a thousand species in regions of the ocean earmarked for industrial activity is an achievable target. High-throughput, integrative taxonomy can, therefore, provide the data that is needed to monitor various ecosystem services (such as the natural history, connectivity, value and function of species) and to help break the regulatory deadlock of high-seas conservation.

**ADRIAN G  GLOVER\*, HELENA WIKLUND, CHONG CHEN AND THOMAS G DAHLGREN**

## Introduction

The growth of industrial activity in our oceans is astonishing. Deep-sea mining, offshore energy, underwater cables, high-seas fisheries and marine biotechnology are just some sectors of the 'blue economy' (*European Commission, 2018*). This economy is targeted for growth to satisfy our ever-increasing appetite for food, energy, technology and wealth. But as we dive deeper into our unexplored oceans to search for new resources and new ideas, there is a growing need to make these efforts sustainable.

To date, major projects associated with a sustainable approach to deep-sea exploitation have, in our opinion, been overly ecosystem-based (*Pikitch et al., 2004*). Attempting a holistic understanding of communities and habitats might work well in a terrestrial environment, where most of the species in a given ecosystem are known. But in our deep oceans, they are not. The vast majority of species have only recently been identified, and most remain undiscovered (*Bouchet et al., 2016*). We contend that

sustainability and growth for the blue economy will only be achieved with an initial 'taxon-focused approach' to conservation. We cannot, essentially, model the ecosystem-wide impacts of deep-sea exploitation without any knowledge of the animals that live there.

Deep-sea mining is an example of a major industrial activity that has been severely constrained by a lack of basic scientific knowledge, even though the cost of obtaining this information would be small relative to the potential size of the industry. Ever since the 1960s, when John Mero first enthused the geological world with a 'call to arms' to reap the riches of the Pacific (manganese, or polymetallic nodules being the primary goal), the industry has been an emerging one, always 'just ten years away' from reality (*Mero, 1965*).

The engineering challenges are considerable, but not fundamentally a problem, and as far back as 1978, riser pipes were used to recover some 800 tonnes of nodules from the

**Competing interests:** The authors declare that no competing interests exist.

remarkable depth of 5,500 m (*Nimmo et al., 2013*). The progress of deep-sea mining since those early explorations has been well-reviewed elsewhere (*Secretariat of the Pacific Community, 2013*). In summary, the constraints on commercial exploitation have been driven by a remarkable global effort to regulate the deep-sea floor, thanks to the United Nations Convention on the Law of the Sea (UNCLOS) and the body that was set up under that legislation – the International Seabed Authority (ISA). Together, UNCLOS and the ISA have, commendably, put the brakes on deep-sea mining until a suitable legal, financial and environmental set of rules that are fair to all has been agreed. And it is for these environmental rules that we require this missing scientific knowledge.

## What animals actually live in regions targeted for industrial activity?

The most active area for industrial seafloor activity in the high seas is a region called the Clarion-Clipperton Zone (CCZ) in the central Pacific

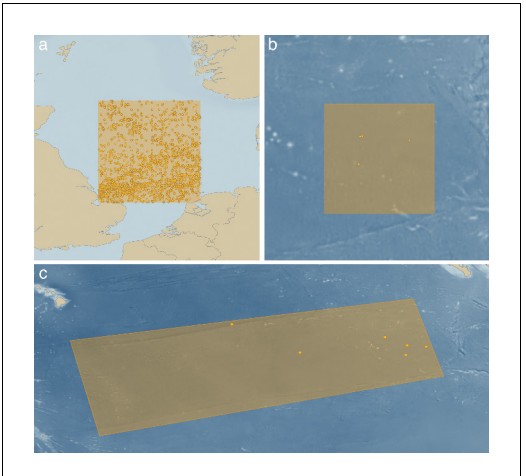

**Figure 1.** Highlighting the absence of faunistic data in deep-sea mining exploration regions using the Ocean Biogeographic Information System. (**a**) A 5° (300,000 km$^2$) search box centered on the shallow North Sea with over 80,000 records from 1,500 annelid worm taxa. (**b**) The same size search box centered on the eastern Clarion-Clipperton Zone with just nine records from five taxa. (**c**) An expanded search box for the entire six million km$^2$ CCZ showing only 12 records (*OBIS, 2018*). Criteria used: Phylum: Annelida, Sample Depth >500 m.

DOI: https://doi.org/10.7554/eLife.41319.002

Ocean. It is a region of approximately six million km$^2$ with typical depths of 4,000-5,500 m and a sedimented, abyssal plain seafloor rich in poly-metallic nodules – small potato-sized mineral accretions that sit on top of, or just slightly beneath, the sediment-water interface. Over 21 billion tonnes of nodules have been estimated to exist in the CCZ; moreover, for nine metals critical to industry (including manganese, cobalt and nickel essential for modern green technologies such as battery-powered cars) the reserves in the CCZ nodules exceed the entire terrestrial reserve base (*Hein et al., 2013*). As of July 2018, 16 exploration contracts with the ISA have been signed for nodule mining in the CCZ, including five with European Union states (Germany, France, Belgium and two with the UK). In recent years, between 10 and 20 exploration cruises have taken place every year to the CCZ (*ISA, 2015*). An extremely conservative estimate for the total number of research expeditions to the CCZ is at least 200, considering that it has been actively explored for the last 40 years (*Glover et al., 2015*).

Given that survey and exploration of the CCZ is at such astonishing levels relative to any other high-seas seafloor ecosystem, it might be expected that such basic things as a list of the animals that live there is relatively easy to obtain. But a simple analysis using the Ocean Biogeographic Information System, the most up-to-date source of taxonomic records in our oceans that pulls data from most major museum collections and databases, shows the complete opposite (*Figure 1*; *OBIS, 2018*). For example, we searched for records of annelid worms (a dominant animal in both shallow and abyssal sedimented habitats) in a five-degree box (~300,000 km$^2$) in the shallow North Sea, recovering over 80,000 records of 1,500 taxa. However, a five-degree box centered over the eastern CCZ (the best-explored part of the CCZ) recovers just nine records from five taxa. Remarkably, even extending the search area to the entire CCZ (~6,000,000 km$^2$) still recovers only three additional records (*Figure 1*).

So, is the CCZ particularly depauperate in annelid worms? It is not, and we know this from the few broad-scale studies of community ecology, which show that the abyssal CCZ is one of the most biodiverse sedimented marine habitats on our planet (*Glover et al., 2002*; *Neal et al., 2011*). Abyssal plain annelid worm biodiversity is

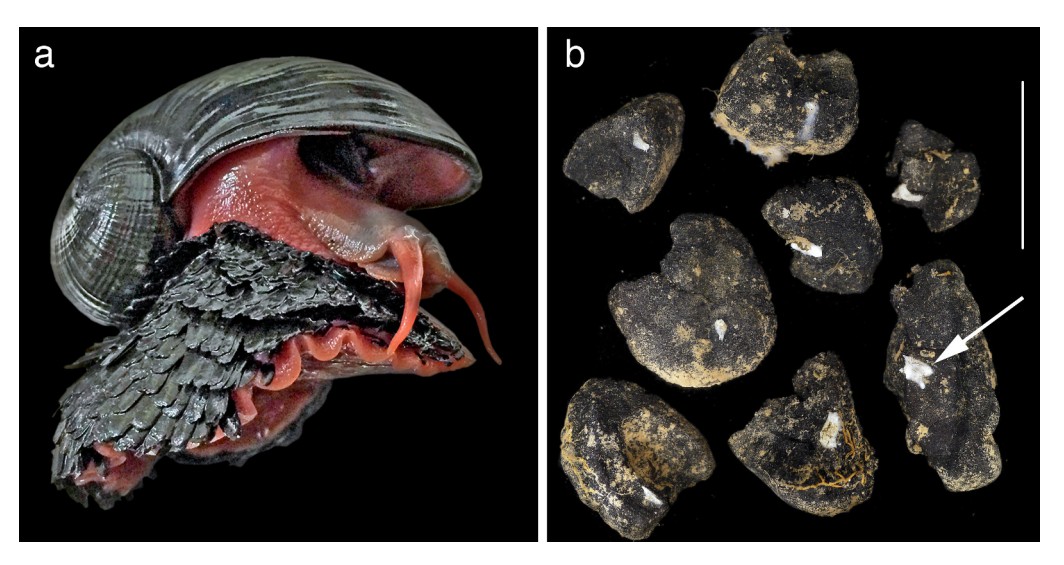

**Figure 2.** Two examples of a taxon-focused approach to conservation in the deep sea that identify both new discoveries of ecosystem services and new approaches to management based on hard evidence. (a) The 'scaly-foot gastropod' *Chrysomallon squamiferum* is the 'signature' taxon discovered at Indian Ocean hydrothermal vents. As the only known animal to use iron in its skeleton, its discovery opens up new biological knowledge and the ability to 'value' environments such as hydrothermal vents (*Chen et al., 2015c*). Shell length 4 cm. (b) The small sponge *Plenaster craigi* is probably the most common animal living on nodules in the Clarion-Clipperton Zone, and was only described in 2017 (*Lim et al., 2017*). It is also a potentially useful monitoring taxon given our new knowledge of its distribution and functional role in filter-feeding on the small potato-sized nodules targeted for deep-sea mining. Scale bar 5 cm.
DOI: https://doi.org/10.7554/eLife.41319.003

typically two to three times the diversity of shallow slope environments along our continental margins (*Neal et al., 2011*). The reason for the lack of faunal lists of the CCZ is that there has been almost no taxonomy done, and no systematic archiving of faunal data with accessible, vouchered and databased material in open, curated collections.

## The historic failure of taxonomy on the high seas

It is worth reflecting briefly on the reasons for this taxonomic failure, before discussing the implications and some potential solutions. There has certainly been no lack of collecting effort; we know that many hundreds of CCZ expeditions have taken place and perhaps tens of thousands of biological samples have been collected (*ISA, 2015*; *Glover et al., 2015*). Almost certainly, the reason is the lack of resources to complete post-expedition analysis and archiving of samples and data. Typically, a deep-sea expedition has a much higher relative cost compared to an inshore or terrestrial operation. A typical 30-day expedition with 20 scientists

and 20 crew to the CCZ on a modern research vessel would cost $1–2m, depending on equipment used and excluding the salary costs of the researchers. Given that most wealthy nations funding large-scale science projects typically work on a three to five year set budget of per year, it is obvious to see that a deep-sea expedition will have a much lower relative budget for post-expedition analysis.

The issue is exacerbated by what is probably best described as a 'fear of the scale of the problem'. Or, the job is so big that nobody actually wants to start it. If there are 10,000 species to describe and archive from the CCZ, and a funded project can only realistically do 100 species, a 1% target may not suffice to impress a grant-awarding body. Furthermore, taxonomic assessment is generally not seen as hypothesis-driven research, and can as such be lower-ranked than other projects (*House of Lords, 2008*). Interestingly, in recent years, almost all taxonomic studies in the CCZ that include full archiving of samples and genetic data have been funded by private contractors. These investors have realized how important baseline

taxonomic data are, and are not perhaps concerned with the trending fashions of ecosystem-based science (*Dahlgren et al., 2016*; *Glover et al., 2016*; *Wiklund et al., 2017*; *Gooday et al., 2018*; *Lim et al., 2017*). Nevertheless, these studies represent informal and formal descriptions of only 54 taxa (none of which are annelids) out of perhaps several thousand left to describe.

## Without faunistic knowledge, there is no knowledge

The father of modern taxonomic nomenclature – Carl Linnaeus – recognized in the 18th century that in order to make better use of biological diversity it was essential to systematize and standardize the naming system of species. The voyaging of Daniel Solander, James Cook, Joseph Banks and others during the Enlightenment was largely driven by a desire to categorize and make use of the world – an early form of a sustainable blue economy. As we explore newly-discovered parts of our planet today, this need remains. However, the modern obsession with the ecosystem-based approach to conservation and sustainability – commendable in regions where we know the components of those ecosystems – does not translate well to the deep oceans where our knowledge is so poor.

It is a central dogma of modern conservation that one of the main reasons we seek to conserve the world around us is for the essential services it provides. Some of these, soil quality or inshore fisheries for example, are quite easy to understand and hence can be used as clear policy frameworks. Translating these types of services into a deep-sea context is challenging, but has been attempted. For example, Thurber et al. list carbon sequestration, methane oxidation, hydrocarbon extraction, deep-sea mining, deep-sea fisheries and many other examples of services provided by the deep seas; they also make a noble attempt to list cultural benefits (such as inspiration and education) that are provided by research into the deep seas (*Thurber et al., 2014*). But combining abiotic services such as carbon sequestration or mineral wealth, which are not essentially reliant on seafloor animals, with biotic ones such as fisheries, leads to a rather confusing conservation message. The truth is that the ecosystem function, let alone service, of most deep-sea animals is completely unknown, as the animals themselves are undiscovered.

We contend that in order to understand the functioning of, and services provided by our deep-ocean ecosystems, we must study the integrative taxonomy and natural history of the animal components of that ecosystem. A fully integrative taxonomy that includes all available information on the biology and traits of new organisms coupled to archived specimens is a clear necessity for the study of ecosystem function and service (*Will et al., 2005*; *Wheeler, 2018*). Such an approach applied to conservation in the deep sea would open up both new discoveries of ecosystem services and new approaches to management, based on hard evidence. We present here two contrasting examples.

In 2001, a remarkable new species of animal called the 'scaly-foot gastropod' was discovered at the hydrothermal vents of the deep Indian Ocean (*Warén et al., 2003*). Now formally known as *Chrysomallon squamiferum* (*Chen et al., 2015a*), this animal is typical of charismatic vent fauna in its evolutionary novelty and remarkable adaptations – the only known metazoan to use iron in its skeleton (*Figure 2a*). The discovery and detailed taxonomic study of this single animal has opened up a vast wealth of new knowledge on both the ecosystem value of hydrothermal vents and the challenges of management in an extraordinarily spatially-constrained and at-risk habitat (*Chen et al., 2015a*; *Chen et al., 2015b*; *Chen et al., 2015c*; *Sigwart et al., 2017*). It is hard to imagine how it would be possible to 'value' the Indian Ocean vent sites without knowledge of this taxon – it is this iconic species that defines the vent in its scientific and public worth.

The second example is the newly-described deep sea sponge *Plenaster craigi* (*Figure 2b*; *Lim et al., 2017*). Contrary to the constrained vent habitat of the scaly-foot gastropod, *P. craigi* lives in what must be the largest continuous and open ecosystem on the planet – the eastern Pacific abyssal plain. Despite over 40 years of intensive sampling in the CCZ, the most abundant animal living on nodules had been completely overlooked. As a slow-growing, widespread filter-feeding sponge with perhaps a rich microbiome with biotechnological spin-offs, it would seem be exactly the sort of animal that one would want to know about before deciding on a management strategy for the region. Yet, it is only now that we have taxonomic information on it, and it is only a single taxon out of so many more.

Faunistic knowledge gained through a taxon focus provides more than an intrinsic value to ecosystems. In the case of *P. craigi*, new microsatellite DNA data demonstrate the potential value of a new protected area to the south of the mining zones (*Taboada et al., 2018*). And more broadly, taxonomic knowledge that includes spectacular images or videos of deep-sea animals is invaluable in connecting broader society to the intrinsic value of our wilderness regions. Ideally, sustainable development should lead to both economic benefits and natural benefits to people – and if people are unaware of what they are conserving, they do not realize the benefit.

## Setting new goals

The ISA has made commendable efforts to regulate industrial activity in the deep sea, and in recent years, a raft of documents and recommendations has been produced, outlining both the process of gaining an exploitation contract (*ISA, 2018*) and a broader regional environmental management plan for some areas, including the CCZ (*ISA, 2011*). But we think that there has been an over-emphasis on a 'holistic' ecosystem approach to environmental data and ecosystem services, when what is most urgently needed are things as basic and fundamental as a list of the species that live in a contracted region, linked to archived samples and genetic material. The universally recognized International Union for Conservation of Nature (IUCN) Red List, for example, is widely used to manage terrestrial ecosystems. But information required for assessment is lacking for the vast majority of the named deep-sea animals, and those undescribed are not even eligible for evaluation.

A reasonable solution would be for the regulator (the ISA) to set targets for the provision of new taxonomic data that includes accessible vouchered material and genetic data. Deep-sea contractors are already funding some taxonomic work. For example, in the period between June 2017 to June 2018, eight peer-reviewed papers on the taxonomy of CCZ animals were published (*Glover, 2018*). These efforts should be recognized and encouraged by the ISA, and perhaps a minimum target set of 100 taxonomic descriptions per year (95 appeared in the period described above). This could be relatively easily achieved with high-throughput, integrative DNA taxonomy approaches (*Glover et al., 2015*) and use of existing quality samples. This would mean that in ten years' time (when some might quip that deep-sea mining will finally start), 1,000 species from the CCZ would be described. With a very modest funding boost that could be shared jointly amongst contractors and stakeholders, this could easily be delivered within five years. Knowledge of a thousand new species would be a revolutionary leap in our understanding of the biodiversity, connectivity, community ecology, ecosystem function and services of this vast region of our planet. Globally, a new emphasis on taxon-focused conservation may be the only way to develop the sustainable blue economy that we all desire.

## Acknowledgements

The authors acknowledge the many colleagues in deep-sea biology who have contributed to informal discussions relevant to this manuscript, in particular Craig R Smith, Daniel Jones, Pedro Martinez Arbizu, Jeff Drazen, Nick Higgs, Diva Amon, Magdalena Georgieva, Julia Sigwart, Sergi Taboada and Andy Gooday. We also thank the reviewers for their helpful comments. The authors are funded by a range of sources that have assisted in developing these ideas including UK Seabed Resources Ltd, the Gordon and Betty Moore Foundation, the European Commission, JPI Oceans, the Mohamed bin Zayed Species Conservation Fund and the Swedish Research Council FORMAS.

**Adrian G Glover** is in the Life Sciences Department, Natural History Museum, London, United Kingdom. a.glover@nhm.ac.uk
iD http://orcid.org/0000-0002-9489-074X

**Helena Wiklund** is in the Life Sciences Department, Natural History Museum, London, United Kingdom.
iD http://orcid.org/0000-0002-8252-3504

**Chong Chen** is in the Japan Agency for Marine-Earth Science and Technology (JAMSTEC), Yokosuka, Japan.
iD http://orcid.org/0000-0002-5035-4021

**Thomas Dahlgren** is in the Department of Marine Sciences, University of Gothenburg, Gothenburg, Sweden, the Gothenburg Global Diversity Centre, Gothenburg, Sweden and the Uni Research Environment, NORCE, Bergen, Norway.
iD https://orcid.org/0000-0001-6854-2031

*Author contributions:* Adrian G Glover, Conceptualization, Funding acquisition, Investigation, Visualization, Methodology, Writing—original draft, Project administration, Writing—review and editing; Helena Wiklund, Investigation, Methodology, Writing—review and editing; Chong Chen, Investigation, Visualization, Methodology, Writing—review and editing; Thomas G Dahlgren, Funding acquisition, Investigation, Methodology, Project administration, Writing—review and

editing; Dr Glover, Dr Wiklund, Dr Chen and Dr Dahlgren jointly developed the ideas behind the manuscript.

*Competing interests:* The authors declare that no competing interests exist.

## Funding

| Funder | Grant reference number | Author |
|---|---|---|
| UK Seabed Resources Ltd | | Adrian G Glover Thomas G Dahlgren |
| European Commission | 603418 | Adrian G Glover Thomas G Dahlgren |
| Gordon and Betty Moore Foundation | | Adrian G Glover Thomas G Dahlgren |
| Mohamed bin Zayed Species Conservation Fund | 182518473 | Chong Chen |
| Svenska Forskningsrådet Formas | 2014-12285-29251-29 | Thomas G Dahlgren |

The funders had no role in study design, data collection and interpretation, or the decision to submit the work for publication.

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
