## [Decision Letter]

Thank you for submitting your article "Managing a sustainable deep-sea 'blue economy' requires knowledge of what actually lives there" to *eLife* for consideration as a Feature Article. Your article has been reviewed by three peer reviewers. The following individuals involved in review of your submission have agreed to reveal their identity: Derek Tittensor (Reviewer #1); Ann Vanreusel (Reviewer #2); Nelia Mestre (Reviewer #3).

We invite you to submit a revised version of the manuscript that addresses the essential revisions listed below.

Summary:

The reviewers find that Glover et al. present an excellent and compelling opinion feature that describes the serious gaps in our deep-sea taxonomic knowledge and expertise. It is very well written and the manuscript provides an excellent series of arguments to support the point the authors want to make: stimulate taxonomic research in the CCZ area. While the 'taxonomic impediment' of the deep ocean is reasonably well-known within the deep-sea scientific community, the authors link it to the rapid ramping-up of the 'blue economy' and how it hinders our attempts to effectively manage such ecosystems. The authors provide clear and correct arguments, which may have a major impact on the future research and management of the area so generating a significant impact. It may also become a highly cited paper for a large and still growing community that is contributing in one way or another to the development of an environmental management plan for the area.

Essential revisions:

1) The authors suggest that ecosystem-based management requires knowledge of species identities and their taxonomic description. It is not clear to me that this is the case. Why could such management not proceed simply with species functioning identified – i.e. is the role that they play in the ecosystem not more important for ensuring management for ecosystem function rather than taxonomic description? If the role of a species, or a functional group of species, can be identified, why do they additionally need to go through the process of formal description? Having rapid automated approaches or coarse taxonomic or morphological/genetic approaches to identifying ecosystem function (e.g. through key traits) would surely be a more rapid way to respond to commercial pressures on the deep-sea?

I am not a taxonomist or geneticist, but it is also not clear why using OTUs (and storing genetic data) cannot take the place of full taxonomic identification? Surely the pressing urgency of growing industrial usage requires creative solutions that can bypass the slow description process? Can the authors not suggest additional novel solutions to be coupled with the centuries-old (though undeniably important) process of taxonomic description?

Perhaps the authors could look into the literature on why species identity is important over and above species traits or functioning for EBM. For example, a partial answer might be that having multiple species fulfilling the same role can provide a buffer against changing conditions, but there is more out there on the benefits of species diversity vs. functional diversity.

The Red List argument given later on in the manuscript is more compelling as a reason for accelerating taxonomic description – but naming species is not enough for Red List assessment. Fairly good knowledge of ranges and population abundance is also required for an assessment of extinction risk. Is this really feasible simply with somewhat more intensive sampling of the CCZ in the next decade, or is it realistically out of reach in the near future?

Improving our taxonomic knowledge of deep-sea ecosystems is clearly a worthwhile cause. But for a robust argument, the 'added value' over new or alternative approaches – or even identifying traits and functioning – needs to be more clearly articulated. Ideally within the context of an integrated approach that couples taxonomy with more rapid-response or precautionary approaches to deep-sea stewardship outlined herein.

2) Major concerns relate to the highlights of the absence of faunistic data, demonstrated by Figure 1, where for CCZ only 5 taxa of annelid worms are registered. However, in the next heading, the authors report a massive effort in the last years where 54 taxa have been described. This is rather confusing to the reader, and raises the questions: How many of these are annelid worms? Only the 5 taxa reported previously? If there are more, why is this missing from the OBIS database? If more taxa of annelid worms in the CCZ is available, the data should either be included in the OBIS maps in Figure 1, or another map should be added to the figure including the taxa, denoting that it was described in the last 2-3 years, or at least a good justification should be made to make it clearer. At least, it would be useful to report the previous example of annelid worms, otherwise, the important point of the message of this article gets a bit lost.

3) In the second paragraph, the authors suggest that with modest funding 1,000 species could be described in 5 years. I think that following this sentence it would be very useful to point if new cruises are still necessary or not, and if not make it clear that there are already enough specimens preserved in museums, etc, just waiting to be described thus emphasising the idea that no extra cruises are needed, as this reassures that indeed with modest funding this is achievable.

---

## [Author Response]

Essential revisions:1) The authors suggest that ecosystem-based management requires knowledge of species identities and their taxonomic description. It is not clear to me that this is the case. Why could such management not proceed simply with species functioning identified – i.e. is the role that they play in the ecosystem not more important for ensuring management for ecosystem function rather than taxonomic description? If the role of a species, or a functional group of species, can be identified, why do they additionally need to go through the process of formal description? Having rapid automated approaches or coarse taxonomic or morphological/genetic approaches to identifying ecosystem function (e.g. through key traits) would surely be a more rapid way to respond to commercial pressures on the deep-sea?I am not a taxonomist or geneticist, but it is also not clear why using OTUs (and storing genetic data) cannot take the place of full taxonomic identification? Surely the pressing urgency of growing industrial usage requires creative solutions that can bypass the slow description process? Can the authors not suggest additional novel solutions to be coupled with the centuries-old (though undeniably important) process of taxonomic description?Perhaps the authors could look into the literature on why species identity is important over and above species traits or functioning for EBM. For example, a partial answer might be that having multiple species fulfilling the same role can provide a buffer against changing conditions, but there is more out there on the benefits of species diversity vs. functional diversity.The Red List argument given later on in the manuscript is more compelling as a reason for accelerating taxonomic description – but naming species is not enough for Red List assessment. Fairly good knowledge of ranges and population abundance is also required for an assessment of extinction risk. Is this really feasible simply with somewhat more intensive sampling of the CCZ in the next decade, or is it realistically out of reach in the near future?Improving our taxonomic knowledge of deep-sea ecosystems is clearly a worthwhile cause. But for a robust argument, the 'added value' over new or alternative approaches – or even identifying traits and functioning – needs to be more clearly articulated. Ideally within the context of an integrated approach that couples taxonomy with more rapid-response or precautionary approaches to deep-sea stewardship outlined herein.

The reviewers have raised some important points that relate to both how taxonomy has changed over the years and the value of taxonomy to other areas of biology, such as ecosystem function. There is rather a long debate in the general biology and philosophy of science literature on this and it is beyond the scope of our short piece to go into great detail. So, what we have done is insert a new sentence and references to two key papers, the first on ‘integrative taxonomy’ by Will et al., 2005, and the second a recent update to this argument by Wheeler, 2018. It is quite hard to express the importance of descriptive, integrative taxonomy better than this quote from the Wheeler, 2018, paper so I reproduce it here:

“The case for descriptive taxonomy. Imagine walking into a gallery of a major art museum and being disappointed to see all the paintings have been hung face-to-the-wall, each with a barcode glued to the reverse of the canvas (Figure 1). Handed a barcode reader, you are told by the docent that you can confidently identify each and every painting. Reading barcodes, you can tell a da Vinci from a de la Tour, but to what end? It is the unique combinations of subject, technique, colors, composition, and the emotions evoked by the works in their totality, that we seek. A taxonomy based on DNA alone has great utilitarian value, but falls far short of its full potential when combined with descriptive taxonomy to reveal in vivid detail the diversity, origins, and history of complex anatomical attributes of species and clades.”

In specific response to our reviewer, they have questioned whether taxonomy is needed if one can identify taxa using OTUs, learn about their traits, their functions and apply that to conservation issues in the deep sea. Our response is that identifying taxa using DNA (the OTU method), describing the traits of those taxa (e.g. morphology, size, fecundity, dispersal ability, feeding mode, symbioses, associations, biological interactions etc.) and working out what role they play in an ecosystem is exactly what integrative and descriptive taxonomy actually is. This is what we are proposing. We call it taxonomy because a name must be applied to these OTUs in order to link all the information (e.g. traits) that we know about them, and they must be linked to a voucher specimen (type) and where the animal was identified with DNA – the sequences and tissue sample so that can be checked if needed. The linkages of the name to the data to the original source information (the specimen) is just standard best scientific practice across any discipline. There is nothing particularly special about it.

In regards to the points raised by the reviewer that we are not pushing forward ‘alternatives’ to taxonomy (e.g. rapid-throughput barcoding, metabarcoding etc.) is that we do not think these are ‘alternatives’ to taxonomy, they are just a new way of doing taxonomy. We are fully behind them, and in fact the recent papers we cite use a rapid-throughput method (described in detail in Glover et al., 2015) that delivers archived and accessible taxonomic information that can be built upon in the future. These are ‘turbo-taxonomy’ approaches based primarily on DNA that deliver the necessary taxonomic data for deep-sea stewardship in a more rapid-way than individual species descriptions.

We have added sentences and references to these comments in the last two sections of the manuscript which we think deal with these useful comments and improve the manuscript considerably.

2) Major concerns relate to the highlights of the absence of faunistic data, demonstrated by Figure 1, where for CCZ only 5 taxa of annelid worms are registered. However, in the next heading, the authors report a massive effort in the last years where 54 taxa have been described. This is rather confusing to the reader, and raises the questions: How many of these are annelid worms? Only the 5 taxa reported previously? If there are more, why is this missing from the OBIS database? If more taxa of annelid worms in the CCZ is available, the data should either be included in the OBIS maps in Figure 1, or another map should be added to the figure including the taxa, denoting that it was described in the last 2-3 years, or at least a good justification should be made to make it clearer. At least, it would be useful to report the previous example of annelid worms, otherwise, the important point of the message of this article gets a bit lost.

We realise this is a bit confusing and we have fixed it. Firstly, we have updated the map in Figure 1C which had excluded a couple of data points (thus addressing the issue of why the distribution of the points in the maps looked a bit different – something picked up by another reviewer). There are now 12 records (not 9) across the whole CCZ for benthic annelids. We have also made it clear that this is for records below 500m in order to remove some surface plankton tows (which account for quite a few records on OBIS). Secondly, in regards the 54 taxa described in the papers listed, none are annelids. We are working furiously on annelid papers, but it’s a huge job and we have not finished yet. We have published only on cnidaria, echinoderms and molluscs so far – those are the 54 taxa. We have clarified this in the text. The data in Figure 1 is correct, as is the 54 taxa mentioned in the text. Thanks for the help here.

3) In the second paragraph, the authors suggest that with modest funding 1,000 species could be described in 5 years. I think that following this sentence it would be very useful to point if new cruises are still necessary or not, and if not make it clear that there are already enough specimens preserved in museums, etc, just waiting to be described thus emphasising the idea that no extra cruises are needed, as this reassures that indeed with modest funding this is achievable.

We have added a sentence here that addresses this and links to the concept above of high-throughput taxonomy and mentions the need for working on existing samples. Some additional sampling is likely to be needed. We think it is precautionary to not rule out additional sampling as particularly for megafauna (large animals) this is still urgently needed. But it is correct that working on existing DNA-fixed samples could deliver perhaps the majority of what is needed. Thanks for the comment.